# TyphoonMLA: A Mixed Naive-Absorb MLA Kernel For Shared Prefix

**Ahmet Caner Yüzügüler, Ahmet Çelik, Jiawei Zhuang & Lukas Cavigelli**
Huawei
Switzerland
{ahmet.yuzuguler,ahmet.celik,zhuangjiawei,lukas.cavigelli}@huawei.com

## Abstract

Multi-Head Latent Attention (MLA) is a recent attention mechanism adopted in state-of-the-art LLMs such as DeepSeek-v3 and Kimi K2. Thanks to its novel formulation, MLA allows two functionally equivalent but computationally distinct kernel implementations: naive and absorb. While the naive kernels (e.g., FlashAttention) are typically preferred in training and prefill for their computational efficiency, existing decoding kernels (e.g., FlashMLA) rely on the absorb method to minimize HBM bandwidth usage. However, the compute-bound nature of the absorb implementations prohibits performance benefits from data reuse opportunities in attention calculations, such as shared prefixes. In this work, we introduce TyphoonMLA, a hybrid approach that combines naive and absorb formulations to harness the strengths of both. TyphoonMLA effectively leverages the shared prefix by applying the naive formulation to the compute-bound parts of attention calculations, while reducing the bandwidth requirements for non-shared parts by using the absorb formulation. As a result, TyphoonMLA improves the throughput of attention calculations in MLA architectures by up to 3× and 3.24× on NPU and GPUs, and boosts end-to-end throughput by up to 1.48× in tokens per second, with only a 3% overhead in HBM size.

## 1 Introduction

Large Language Models (LLMs) have been widely adopted in various application domains, ranging from chat assistants (OpenAI, 2023) to coding agents (Chen et al., 2021), thanks to their unprecedented language processing and reasoning capabilities. However, their substantial computational requirements result in slow and inefficient inference, which undermines the user experience and increases operational costs. Therefore, addressing these computational challenges is essential for enabling broader adoption and ensuring their sustainable deployment.

To improve the efficiency of LLM inference, a new attention architecture called Multi-Head Latent Attention (MLA) has recently been introduced (DeepSeek-AI et al., 2024a). In this architecture, the key and value tensors that hold contextual information from previous tokens, i.e., the KV-cache, are stored in a low-rank latent space, which helps eliminate memory bandwidth bottlenecks in attention layers. MLA's flexibility to merge the latent-space projection layers with matrix multiplication operations in attention calculations (i.e., *absorption* trick) makes it possible to implement MLA in two functionally equivalent but computationally distinct ways, namely, **naive** and **absorb**.

In training and prefill, where performance is typically limited by the compute capacity (e.g., the total throughput of Tensor cores in GPUs), the compute-efficient naive implementation is preferred. In decode, where the HBM bandwidth dictates the performance, the memory-efficient absorb implementation is used. As a result, MLA architectures utilize computational resources more effectively both in training and inference, making them a more efficient alternative to other attention architectures, such as Multi-Head Attention (MHA) (Vaswani et al., 2017) or Grouped-Query Attention (GQA) (Ainslie et al., 2023). Consequently, MLA serves as the backbone of various state-of-the-art LLMs, such as DeepSeek-v3 (DeepSeek-AI et al., 2024b) and Kimi K2 (Bai et al., 2025).

Another promising direction to improve the efficiency of attention kernels is prefix sharing (Juravsky et al., 2024). In various application scenarios, a portion of the KV-cache is shared across multiple

queries, creating a data reuse opportunity that can help alleviate memory bottlenecks. For instance, many of the inference services today employ a system prompt, which is shared among all user queries to enhance the safety and quality of generated responses. Both official documentation and recent leaks through prompt-injection attacks have revealed that system prompts used in popular inference services reach lengths of tens of thousands of tokens (e.g., Claude-4 has a system prompt of 26k tokens (Johnson, 2025)). Furthermore, parallel reasoning techniques, such as Tree-of-thought (ToT) (Yao et al., 2023) and Graph-of-thought (GoT) (Besta et al., 2024), as well as the speculative decoding techniques (Wang et al., 2025a), often result in multiple queries attending to the same part of the KV-cache, boosting the potential for data reuse in attention computation.

To effectively benefit from this potential data reuse in attention calculations, recent studies have proposed several MHA and GQA kernel implementations (Yao et al., 2025; Pan et al., 2025; Wang et al., 2025b). In these designs, the shared parts of the KV-cache are read only once from HBM and reused across multiple queries, reducing HBM accesses and mitigating memory bandwidth bottlenecks. Although these techniques improve the efficiency and performance of MHA and GQA kernels, they are not directly applicable to MLA, since the performance of a typical MLA kernel implementation is limited by computation rather than memory bandwidth. As a result, current MLA kernels fail to fully exploit the data reuse opportunities available in attention computation.

In this paper, we propose a novel MLA method that effectively leverages data reuse in the shared parts of the KV-cache to improve the efficiency and performance of attention computation. Our key observation that leads to the proposed solution lies in the fact that, while the absorb implementation is preferred in memory-bound regions, the naive implementation becomes more efficient in the compute-bound regions when the KV-cache is shared across multiple queries, as it requires fewer floating-point operations than the absorb implementation. Building on this insight, our proposed MLA method merges the naive and absorb implementations: it uses the naive formulation in the shared parts of the KV-cache to exploit its computational efficiency and the absorb formulation in the non-shared parts of the KV-cache to benefit from its memory bandwidth efficiency.

We demonstrate through extensive experiments on GPUs and NPUs that TyphoonMLA substantially improves resource utilization and offers a speedup of up to $3.2\times$. Moreover, the proposed method is compatible with other optimization techniques, such as PagedAttention (Kwon et al., 2023) and RadixAttention (Zheng et al., 2024), and it supports existing parallelization strategies, such as tensor and sequence parallelism. Therefore, TyphoonMLA can be effortlessly integrated into popular inference frameworks, such as vLLM and SGLang. Furthermore, the proposed method is mathematically equivalent to standard MLA implementations; thus, it does not cause any accuracy degradation and requires neither training nor fine-tuning.

In short, this paper makes the following contributions:

- We introduce TyphoonMLA, a novel MLA inference method that achieves higher computational efficiency and resource utilization in attention calculations. To the best of our knowledge, this is the first work that combines both naive and absorb implementations in MLA calculations.

- We provide a detailed analysis that shows that TyphoonMLA requires fewer floating-point operations in compute-bound regions and consumes less HBM bandwidth in the memory-bound regions than existing MLA kernels.

- We conduct a series of experiments on NPUs and GPUs, demonstrating that TyphoonMLA improves attention throughput by up to $3.2\times$ for DeepSeek-v3 and Kimi K2, with only a 3% overhead in memory footprint.

Our code is open-sourced and publicly available[1]. Generative AI tools were used to edit and refine this paper to improve the clarity and quality of writing.

---

[1] https://github.com/huawei-csl/TyphoonMLA-community

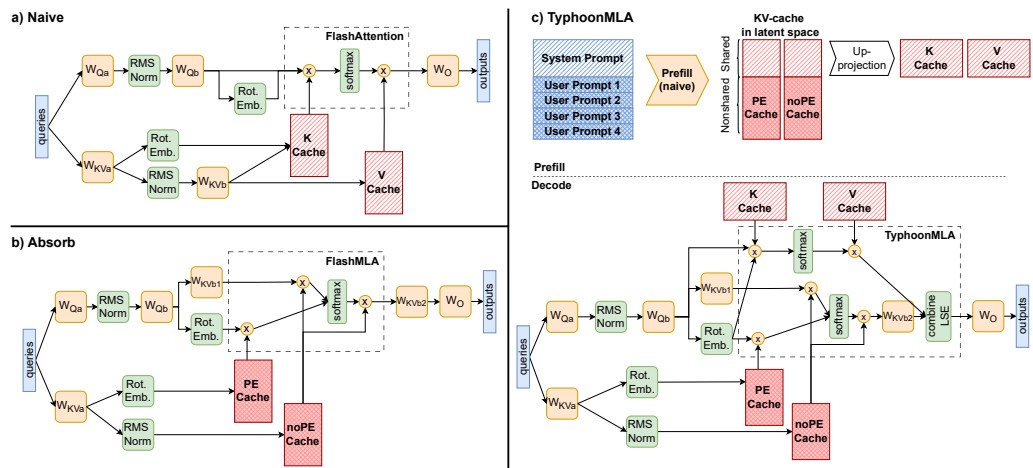

Figure 1: a) The naive formulation of MLA. b) The absorb formulation of MLA. c) The prefill and decode stages of TyphoonMLA.

## 2 BACKGROUND

In this section, we first provide background on MLA and describe its two implementation variants: naive and absorb. We then discuss prefix sharing in LLM inference and its potential for improving the efficiency of self-attention kernels in MLA layers.

### 2.1 MULTI-HEAD LATENT ATTENTION

MLA is an attention mechanism introduced by DeepSeek-v2 and adopted by state-of-the-art LLMs such as DeepSeek-v3 (DeepSeek-AI et al., 2024b), Kimi K2 (Bai et al., 2025), and Long-Cat (Meituan et al., 2025). Unlike standard attention mechanisms, MLA employs a low-rank key-value joint compression mechanism to reduce KV-cache size and bandwidth requirements during inference. Moreover, its novel positional encoding enables the key-value projections to be rearranged in a way that leads to two computationally distinct implementations, namely naive and absorb.

**Naive:** The naive implementation of MLA keeps the KV-cache in an uncompressed form. Fig. 1(a) illustrates the components of MLA in a naive implementation. Therein, after passing through the rotational embedding and RMS normalization layers, the key and value tensors are decompressed using the up-projection matrix, denoted as $W_{KVb}$ in the figure. As a result, the K and V caches consist of multiple attention heads, and the self-attention calculation becomes equivalent to the standard MHA formulation.

**Absorb:** The absorb implementation of MLA utilizes the commutative property of matrix multiplication to reposition the up-scaling matrix. In absorb formulation, as shown in Fig. 1(b), the up-projection matrix is split into two submatrices (denoted as $W_{KVb1}$ and $W_{KVb2}$) and placed before and after the query ($W_{Qb}$) and output ($W_O$) projection layers. This formulation allows the KV-cache to remain in a compressed form, represented as PE and noPE caches in the figure. However, applying the up-projection matrix to the queries further expands their embedding dimensions, requiring more floating-point operations in the self-attention calculations. Moreover, the resulting formulation is incompatible with existing self-attention kernels (e.g., FlashAttention (Dao et al., 2022)), necessitating the development of specialized kernels for MLA layers (e.g., FlashMLA (Jiashi Li, 2025), ThunderMLA (Spector et al., 2025)).

### 2.2 PREFIX SHARING

In various inference scenarios, large portions of the KV-cache are shared across multiple queries, creating opportunities for data reuse. First, modern inference services often employ system prompts

to enhance the safety and quality of generated responses. These system prompts include tool usage instructions, API call descriptions, and even recent news updates after the model's knowledge cutoff dates, and can consist of tens of thousands of tokens (Johnson, 2025). Second, recent LLM reasoning techniques, such as Chain-of-Thought with self-consistency (Wang et al., 2023), Tree-of-Thought (Yao et al., 2023), and Graph-of-Thought (Besta et al., 2024), reformulate inference as a tree or graph search, in which multiple branches run in parallel while sharing a common prefix. Finally, certain speculative decoding techniques (Wang et al., 2025a) require validating multiple candidate tokens in parallel, which share a long sequence of past tokens. In all these cases, multiple queries attend to overlapping regions of the KV-cache, offering a significant data reuse opportunity to improve the efficiency and performance of attention computations.

## 3 PROPOSED METHOD

In the previous section, we discussed two distinct ways of implementing MLA. In this section, we provide details about TyphoonMLA, which combines both implementations.

### 3.1 TYPHOONMLA

As previously discussed, the shared prefix introduces data reuse in the shared parts of the attention calculations. Building on this insight, TyphoonMLA partitions the attention calculation into low- and high-arithmetic-intensity components and applies absorb and naive formulations, respectively. Fig. 1(c) illustrates how TyphoonMLA works.

**Prefill:** In the prefill stage, a tree-like query structure is formed from the incoming user requests and a shared prefix (e.g., a system prompt). The queries are processed by the LLM using a prefix-aware naive kernel, producing PE and noPE caches in the low-rank latent space. The portion of the PE and noPE caches that corresponds to the shared prefix is then expanded via the up-projection to form the K and V caches. Consequently, the shared and non-shared parts of the KV-cache are stored in uncompressed and compressed formats, respectively, enabling the use of both naive and absorb formulations. Notably, the up-projection incurs no additional computational overhead, as it is already performed by standard naive kernels during the prefill stage.

**Decode:** In the decode stage, TyphoonMLA first applies the down-projection layers $W_{Qa}$ and $W_{KVa}$ to the queries, obtaining the $q$ and $kv$ vectors. The $q$ vectors pass through an RMS normalization layer, followed by the up-projection layer $W_{Qb}$. The $kv$ vectors are split into two subvectors, one of which passes through a rotational embedding layer, and the other through an RMS normalization. All operations up to this point are common to TyphoonMLA, naive, and absorb implementations.

Unlike naive and absorb implementations, TyphoonMLA performs self-attention calculations using both compressed and uncompressed KV-cache while reusing the shared operations. As described in Algorithm 1, TyphoonMLA takes the output of $W_{Qb}$ up-projection layer ($Q$), shared KV-cache in uncompressed form ($C_K$ and $C_V$), non-shared KV-cache in latent form ($C_N$ and $C_R$), and KV up-projection matrices ($W_{KVb1}$ and $W_{KVb2}$) as inputs. The input queries $Q$ are first split from their

---

**Algorithm 1** TyphoonMLA Decode Attention Kernel

---

**Require:** Queries after $\mathbf{W}_{Qb}$ proj. and RoPE layers, $\mathbf{Q} \in \mathbb{R}^{B \times H \times D_{qk}}$
**Require:** Shared KV-cache in uncompressed form, $\mathbf{C}_K \in \mathbb{R}^{B \times L_s \times D_{qk}}$, $\mathbf{C}_V \in \mathbb{R}^{B \times L_s \times D_v}$
**Require:** Non-shared KV-cache in latent space, $\mathbf{C}_N \in \mathbb{R}^{B \times L_n \times D_l}$, $\mathbf{C}_R \in \mathbb{R}^{B \times L_n \times D_r}$
**Require:** KV up-projection matrices, $\mathbf{W}_{KVb1} \in \mathbb{R}^{H \times D_n \times D_l}$, $\mathbf{W}_{KVb2} \in \mathbb{R}^{H \times D_v \times D_l}$

---

1: $[\mathbf{Q}_N, \mathbf{Q}_R] \leftarrow \mathbf{Q}$          ▷ Split $\mathbf{Q}$ from dimension $D_{qk}$ into $[D_n, D_r]$
2: $\mathbf{Q}_R \leftarrow \text{RoPE}(\mathbf{Q}_R)$          ▷ Apply positional encoding
3: $\mathbf{Q}_K \leftarrow [\mathbf{Q}_N, \mathbf{Q}_R]$          ▷ Merge after RoPE
4: $\mathbf{O}_N, l_N \leftarrow \text{softmax}(\mathbf{Q}_K \mathbf{C}_K^\mathsf{T}) \mathbf{C}_V$          ▷ Compute naive component
5: $\mathbf{Q}_A \leftarrow \mathbf{Q}_N \mathbf{W}_{KVb1}$
6: $\mathbf{O}_A, l_A \leftarrow \text{softmax}(\mathbf{Q}_A \mathbf{C}_N^\mathsf{T} + \mathbf{Q}_R \mathbf{C}_R^\mathsf{T}) \mathbf{C}_N$          ▷ Compute absorb component
7: $\mathbf{O}_A \leftarrow \mathbf{O}_A \mathbf{W}_{KVb2}^\mathsf{T}$
8: $\mathbf{O} \leftarrow \text{combine}(\mathbf{O}_N, \mathbf{O}_A, l_N, l_A)$          ▷ Combine partial outputs using LSEs
9: **return O**

---

Table 1: Computational analysis of naive, absorb, and TyphoonMLA. $B$: batch size, $S_q$: query sequence length, $L_s$: shared context length, $L_n$: non-shared context length, $H$: number of heads, $D_{qk}$: head dim. for Q and K, $D_v$: head dim for V, $D_l$: KV LoRA rank, $D_n$: noPE head dim, $D_r$: RoPE head dim. TyphoonMLA always requires smaller memory operations than naive (highlighted in red) and fewer multiply-accumulate operations (MACs) than absorb (highlighted in blue).

| | **MAC** | DEEPSEEK-V3 (×1024) |
|---|---|---|
| NAIVE | $BS_qL_sH(D_{qk}+D_v) + BS_qL_nH(D_{qk}+D_v)$ | $40 \times BL_s + 40 \times BL_n$ |
| ABSORB | $BS_qL_sH(2D_l+D_r) + BS_qL_nH(2D_l+D_r)$ | $136 \times BL_s + 136 \times BL_n$ |
| TYPHOONMLA | $BS_qL_sH(D_{qk}+D_v) + BS_qL_nH(2D_l+D_r)$ | $40 \times BL_s + 136 \times BL_n$ |
| | **HBM R/W** | DEEPSEEK-V3 (×1024) |
| NAIVE | $L_sH(D_{qk}+D_v) + BL_nH(D_{qk}+D_v)$ | $40 \times L_s + 40 \times BL_n$ |
| ABSORB | $L_s(D_l+D_r) + BL_n(D_l+D_r)$ | $0.56 \times L_s + 0.56 \times BL_n$ |
| TYPHOONMLA | $L_sH(D_{qk}+D_v) + BL_n(D_l+D_r)$ | $40 \times L_s + 0.56 \times BL_n$ |

embedding dimension into two tensors, and a positional encoding is applied. Then, similar to the naive implementation, the queries are multiplied by the uncompressed K cache, $C_K$, passed through a softmax function, and multiplied by the V cache, $C_V$. Similar to the absorb implementation, the queries are first up-projected using the $W_{KVb1}$ matrix, multiplied by the compressed PE and noPE caches ($C_R$ and $C_N$), passed through a softmax function, and multiplied again by the noPE cache. As required by the absorb formulation, the output of the absorb component is then up-projected once again using the $W_{KVb2}$ matrix. Finally, the partial results from these two components are then aggregated using a *combine* function with the help of the log-sum-exp (LSE) of the softmax denominators ($l_N$ and $l_A$). TyphoonMLA is mathematically equivalent to both absorb and naive implementations; therefore, it does not require any re-training or fine-tuning.

**Fall-back to Absorb:** At small batch sizes, when there is insufficient data reuse, the naive implementation can become less efficient than absorb due to higher memory access costs. To address this, TyphoonMLA automatically switches to an absorb-only kernel whenever the batch size falls below a predefined threshold. By doing so, TyphoonMLA avoids any performance penalty at small batch sizes, ensuring consistently high efficiency across a wide range of batch sizes.

**Parallelization:** TyphoonMLA is fully compatible with existing parallelization strategies for attention computation. Although the compressed PE and noPE caches have a single head dimension, the uncompressed K and V caches can be parallelized across attention heads. Additionally, both compressed and uncompressed caches can be easily parallelized across the sequence dimension. As a result, TyphoonMLA seamlessly supports both tensor and sequence parallelism, enabling efficient scaling across multiple NPUs or GPUs.

## 3.2 COMPUTATIONAL ANALYSIS

To quantify the potential performance benefits of the proposed method, we derive the computational requirements of attention calculations in terms of the number of multiply-accumulate operations (MAC) and the size of memory read/write operations (HBM R/W) for TyphoonMLA, as well as the naive and absorb formulations. Table 1 summarizes these derivations with respect to architectural and generation parameters. In the right-most column, we substitute the architectural parameters with those of the DeepSeek-v3 to facilitate a direct comparison. In our analysis, we consider only self-attention computations, excluding the projection layers, since self-attention dominates execution time at long sequence lengths and asymptotically approaches the total runtime.

Since the absorb implementation is typically compute-bound, its performance is primarily determined by the number of MACs. In contrast, the naive implementation is generally memory-bound, so its efficiency is dominated by the amount of data read from HBM. Therefore, to maximize overall performance, TyphoonMLA aims to achieve lower MACs than absorb in compute-bound regions and fewer HBM read and writes than naive in memory-bound regions.

Table 1 shows that the naive implementation requires reading $(40 \times L_s + 40 \times BL_n) \times 1024$ words from HBM. In comparison, TyphoonMLA reads only $(40 \times L_s + 0.56 \times BL_n) \times 1024$ words, which is the same for the shared portion but approximately 70× smaller for the non-shared portion. The absorb implementation, typically compute-bound, requires $(136 \times BL_s + 136 \times BL_n) \times 1024$ MAC operations. TyphoonMLA, in contrast, requires only $(40 \times BL_s + 136 \times BL_n) \times 1024$ MAC operations, which is the same for the non-shared portion, but 3.4× smaller in the shared portion. This analysis reveals that TyphoonMLA requires **fewer bytes** to read from HBM than the naive formulation in memory-bound regions and **fewer MACs** than the absorb formulation in compute-bound regions, reducing the overall computational complexity of attention calculations in MLA.

To combine the partial results of the naive and absorb parts, TyphoonMLA uses a *CombineLSE* function, similar to the epilogue stage in Flash Attention (Dao et al., 2022). This function performs only vector operations and requires reading $2BS_qHD_v$ bytes from HBM and performing $2BS_qHD_v$ MAC operations. Since the computational complexity of this function is independent of the KV sequence length, which typically ranges from hundreds to thousands, its computational overhead is insignificant compared to the other components of the attention calculations.

For TyphoonMLA to achieve a speedup, reading the shared portion of the KV-cache should be faster with the naive implementation than computation with the absorb implementation. This happens only when there is sufficient data reuse, which occurs when the batch size is larger than a threshold. By equating the memory read time of the naive and the computation time of the absorb implementation for the shared part of the KV-cache, we identify the batch size threshold, $B_\theta$, in terms of architectural parameters (defined above) and hardware specifications, namely throughput $T$ and HBM memory bandwidth $M$, as follows:

$$\frac{L_s H(D_{qk} + D_v)}{M} = \frac{B_\theta S_q L_s H(2D_l + D_r)}{T} \implies B_\theta = \frac{(D_{qk} + D_v)}{S_q(2D_l + D_r)}\frac{T}{M} \qquad (1)$$

When the architectural parameters are replaced with those of DeepSeek-v3 and the hardware parameters with Ascend NPU ($T = 376$ TOPS/s, $M = 1.8$ TB/s), we obtain $B_\theta = 61$. This threshold indicates the break-even point, beyond which TyphoonMLA becomes computationally more efficient than the absorb implementation, whereas, for batch sizes smaller than $B_\theta$, TyphoonMLA falls back to the absorb kernel to avoid any slowdown.

## 4 EXPERIMENTS

Building on the complexity analysis presented in Section 3.2, we now provide empirical evidence to validate the performance benefits of TyphoonMLA. In this section, we evaluate the performance of TyphoonMLA and compare it against various baselines on NPUs and GPUs. First, we measure TyphoonMLA's throughput on popular datasets using DeepSeek-v3 and Kimi K2 with different system prompts. Next, we examine its performance breakdown to validate our insights. Finally, we assess TyphoonMLA's memory footprint and compare it against the absorb baseline.

**Experiments on NPUs:** We implemented TyphoonMLA for Ascend NPUs using the CANN toolkit (Huawei, 2024a). For the absorb component of TyphoonMLA, we developed a custom kernel that supports paged and variable-length KV-cache using the Ascend CATLASS library (Huawei, 2025). For the naive component of TyphoonMLA, we employed the `NpuFusedInferAttentionScore` function from TorchNPU (Huawei, 2024b). For the projection layers and combine logic, we used TorchNPU's `einsum` and vector operators.

To evaluate the impact of system prompt length on performance, we selected three system prompts with varying lengths, summarized in Table 2. As benchmark datasets, we used the MMLU (Hendrycks et al., 2021), GSM8K (Cobbe et al., 2021), and SimpleQA (Wei et al., 2024), which comprise questions and answers from various topics. We repeat each experiment for batch sizes of 64, 128, 256, 512, and 1024.

To simulate a realistic deployment scenario, we adopted continuous batching with a paged KV-cache with a block size of 128. Each experiment starts by randomly sampling questions from a dataset and forming a batch of queries. At the end of each decode iteration, the completed queries are replaced with new questions sampled from the dataset. Each experiment is continued until the entire dataset

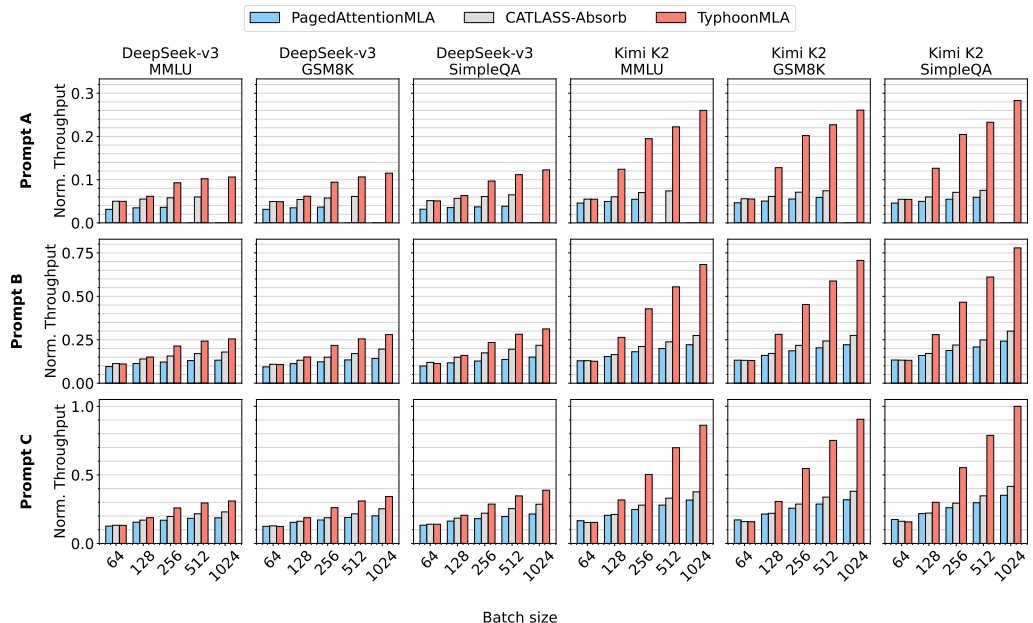

Figure 2: Benchmark results on Ascend NPUs. Y-axes represent normalized throughput in terms of the number of generated tokens per second per layer. Some data points for baselines are missing as their memory footprint exceeds the HBM capacity.

is processed. The throughput is then calculated by dividing the total number of generated tokens by the total execution time across all decode iterations. We calculate the speedup as the throughput of TyphoonMLA divided by the best of the two baselines.

We run the experiments on an Ascend NPU, equipped with 24 Davinci cores and 64 GB of HBM memory, which provides a throughput of 376 TOPS/s in FP16 precision and an HBM bandwidth of 1.8TB/s. We compared TyphoonMLA's throughput with that of the baseline kernels, namely TorchNPU PagedAttentionMLA kernel and a custom-built CATLASS Absorb-only kernel.

Fig. 2 presents the normalized throughput of Typhoon-MLA and the baseline methods for the attention layers of DeepSeek-v3 and Kimi K2. TyphoonMLA consistently outperforms both baselines across all datasets, sys-

Table 2: System prompts used in the experiments, taken from (Johnson, 2025).

| Name | LLM service | #tokens |
|------|-------------|---------|
| Prompt A | Claude-4 | 26472 |
| Prompt B | OpenAI/o3 | 7069 |
| Prompt C | Grok/Personas | 4759 |

tem prompts, and batch sizes, achieving speedups between 1.2× and 3×. As expected, TyphoonMLA offers the highest speedup with Prompt A, since longer system prompts increase the ratio of shared to non-shared portions of the KV-cache. We further observe that the speedup is generally higher for the Kimi K2 than for DeepSeek-v3. This is because the number of attention heads in Kimi K2 is 64, half of DeepSeek-v3, which makes the former's performance more sensitive to data reuse. Overall, these results confirm that TyphoonMLA effectively exploits the shared prefix and delivers consistent and significant speedups over the existing MLA kernels.

**Experiments on GPUs:** We also implemented TyphoonMLA using the FlashInfer naive and absorb MLA kernels (Ye et al., 2025) and repeated our NPU experiments on a GPU. Fig. 3 reports the normalized throughput of TyphoonMLA on a GPU with a 1 PetaFLOPS/s of theoretical throughput in FP16, 3.3 TB/s of HBM bandwidth, comparing it against the FlashMLA (Jiashi Li, 2025) and FlashInfer (absorb-only) baselines at batch sizes of 64, 128, 256, 512, and 1024. TyphoonMLA achieves higher throughputs than both baselines, with factors up to 3.24×. Similar to the NPU results, TyphoonMLA accelerates Kimi K2 more than DeepSeek-v3, owing to its smaller number of

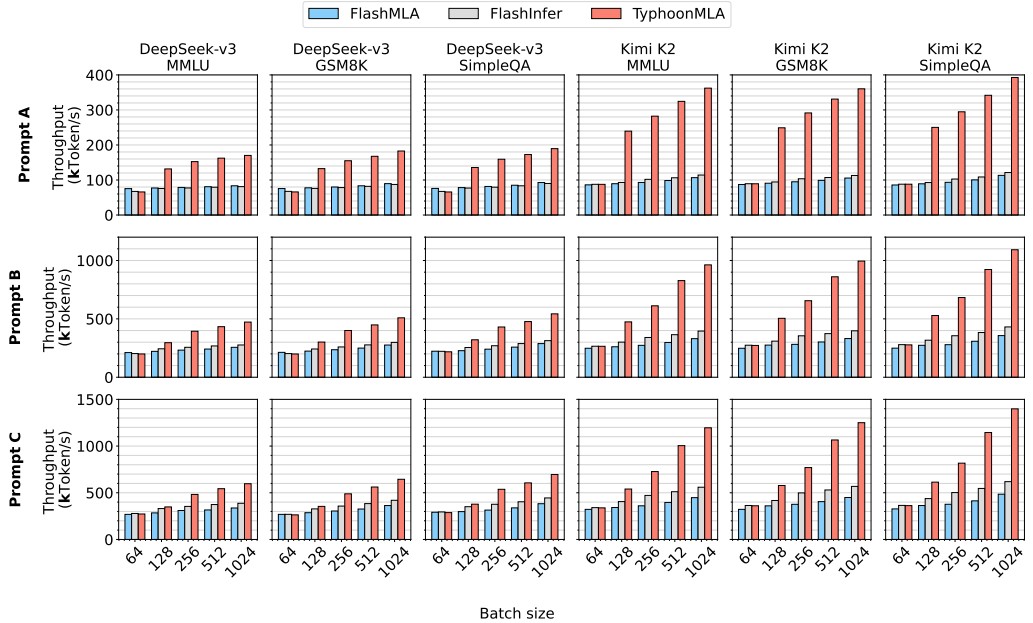

Figure 3: Benchmark results on GPU for various batch sizes. Y-axes represent throughput in terms of the number of generated tokens per second per layer.

Table 3: Token generation rate for DeepSeek-v3 processing MMLU dataset with a batch size of 128 per GPU.

| | FlashMLA | | | TyphoonMLA | | |
|---|---|---|---|---|---|---|
| | Attention time (ms) | Total time (ms) | TGR (kToken/s) | Attention time (ms) | Total time (ms) | TGR (kToken/s) |
| Prompt A | 99.1 | 127.2 | 1.01 | 58.1 | 86.3 | 1.48 |
| Prompt B | 34.5 | 62.6 | 2.04 | 25.9 | 54.0 | 2.37 |
| Prompt C | 26.9 | 55.0 | 2.33 | 22.0 | 50.1 | 2.56 |

attention heads. Moreover, it achieves the highest speedups with Prompt A, as it has the highest token count. Overall, these results demonstrate that TyphoonMLA generalizes effectively to GPUs, delivering substantial performance gains across diverse hardware platforms.

**Latency Breakdown:** To demonstrate how TyphoonMLA accelerates attention computations and how its components behave under different batch sizes, we profiled its execution using the CANN toolkit's `msprof` tool and compared it against the CATLASS absorb-only baseline. Fig. 4 shows the execution time breakdown of TyphoonMLA (bars on the left-hand side) and the CATLASS baseline (bars on the right-hand side). In the figure, we denote the naive and absorb parts of attention computation in TyphoonMLA as *Stage 1, Attn* and *Stage 2 Attn*, respectively; the up and down-scaling required in the absorb formulation as $W_{KV,B1}$-proj and $W_{KV,B2}$-proj; and the combination logic that merges the intermediate results of both stages as *CombineLSE*. We excluded the other linear layers of the attention block, since they are identical across both methods and negligible at long sequence lengths. In this experiment, we set the shared prefix length to 4096, the non-shared sequence length of each request to 512, and used the architectural parameters of Kimi K2. Batch sizes smaller than 128 are omitted, as in these cases, TyphoonMLA falls back to the absorb-only implementation, becoming identical to the baseline. As the CATLASS baseline is absorb-only, it contains only Stage 2 components.

The runtime breakdown of TyphoonMLA shows that its performance gains align closely with the estimates from our theoretical analysis in Section 3.2. At a batch size of 1024, the attention calcu-

lations in the CATLASS baseline take 6.43 ms, while the naive and absorb parts of TyphoonMLA take 1.63 ms and 1.06 ms, respectively. Since the non-shared part of attention is identical between TyphoonMLA and the baseline, we estimate the execution time of the shared part in the baseline as $6.43 - 1.06 = 5.37$ ms. The ratio between the execution times of the shared parts in the baseline and TyphoonMLA is therefore $5.37/1.63 = 3.3$, which matches the reduction in the number of operations between the naive and absorb formulations derived in Section 3.2. These results not only validate our initial insights and theoretical analysis but also demonstrate that TyphoonMLA's hybrid design translates directly into measurable runtime improvements.

**End-to-End Speedup:** Previous experimental results demonstrated that TyphoonMLA significantly accelerates the attention layers in MLA architectures. However, the execution time of the other layers in LLMs (e.g., MoE, collective communication etc.) would remain unchanged. Therefore, we now investigate the impact of the proposed method on the end-to-end LLM decoding performance. To that end, we estimate the *token generation rate* (TGR) for DeepSeek-v3 with a batch size of 128 per device on a system with 128 GPUs by measuring attention layers on a GPU and using the profiling data for other layers provided by DeepSeek-AI (Deepseek-AI, 2025).

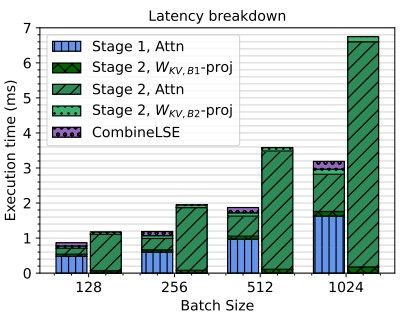

Table 3 reports the estimated TGR and per-iteration execution time for both the attention layers and the full model when using TyphoonMLA, comparing against those with FlashMLA as the baseline. Since attention computation constitutes a substantial portion of overall runtime, the speedup achieved by TyphoonMLA in the attention layers translates directly into notable end-to-end throughput gains, reaching up to a 1.48× improvement in tokens per second.

Figure 4: Latency breakdown of TyphoonMLA (bars on the left-hand side) and CATLASS absorb-only baseline (bars on the right-hand side) for Kimi K2 architecture. Stage 1 and Stage 2 represent the naive and absorb parts of TyphoonMLA, and have sequence lengths of 4096 and 512, respectively.

**HBM Footprint:** Since TyphoonMLA stores the shared portion of the KV-cache in an uncompressed form, its memory footprint differs from that of the absorb baselines. To quantify the impact of TyphoonMLA on the memory footprint, we analyze TyphoonMLA's HBM usage under various deployment settings and compare it against the absorb baseline. We assume that the model is distributed across 384 NPUs, as in a CloudMatrix cluster (Zuo et al., 2025), employing full expert parallelism on MoE layers and a combination of data, tensor, and sequence parallelism of factors 24, 4, and 4, respectively. Fig. 5 shows the HBM size of DeepSeek-v3 for batch sizes ranging from 4K to 32K and maximum sequence lengths from 32K to 256K, assuming Prompt A as the shared prefix.

At small batch sizes and sequence lengths, HBM usage is dominated by the model weights, which are identical for both TyphoonMLA and the absorb baseline. As batch size and sequence length increase, the KV-cache grows for both methods. However, at large batch sizes and sequence lengths, the memory occupied by the KV-cache that corresponds to the shared prefix becomes negligible compared to the non-shared portion, which is identical for both methods. Consequently, TyphoonMLA incurs only a minimal HBM overhead, limited to approximately 3% across a wide range of deployment scenarios.

## 5 RELATED WORK

**LLM Serving Acceleration:** Various prior work proposes system-level optimizations for LLM serving. Orca (Yu et al., 2022) introduced *continuous batching*, which allows replacing the completed requests in a batch with new ones to improve effective throughput. Inspired by the OS virtual memory concept, PagedAttention (Kwon et al., 2023) partitions the KV-cache into memory pages to efficiently handle KV-cache with variable length. TyphoonMLA supports both continuous batching and PagedAttention; thus, it benefits from the performance gains of such system-level optimizations. Furthermore, to distribute LLM inference across multiple devices, existing systems typically

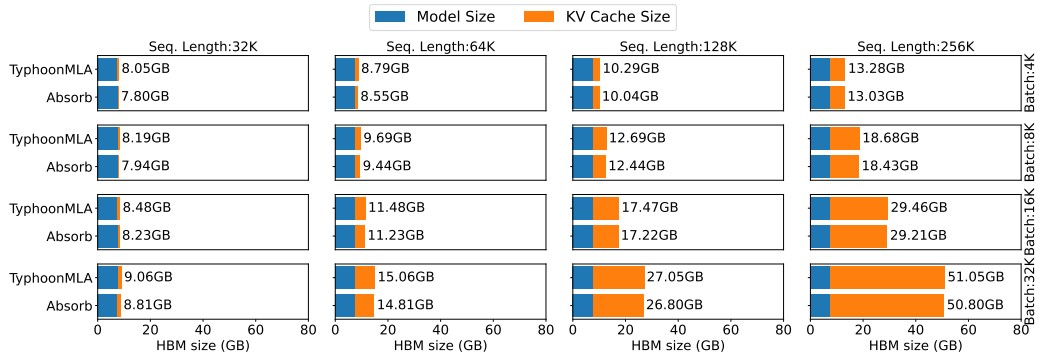

Figure 5: HBM footprint comparison for DeepSeek-v3 in FP8 precision for both weights and KV-cache.

employ tensor (Shoeybi et al., 2019) and sequence (Jacobs et al., 2023) parallelism. TyphoonMLA also seamlessly supports both types of parallelization, enabling deployment at scale.

**Efficient Attention Kernels:** As attention computation constitutes a significant portion of the total execution time of LLM inference, many prior studies focused on developing efficient attention kernels. Flash Attention (Dao et al., 2022) partitions attention computations into tiles that fit on the on-chip buffers to minimize HBM bandwidth usage. FlashDecoding (Dao, 2024) introduces the *Split-K* method to improve the parallelization across GPU cores. Flashinfer (Ye et al., 2025) provides a customizable attention template to facilitate the development of custom attention kernels. However, all these techniques are developed for MHA; hence, they are not applicable to MLA. FlashMLA (Ji-ashi Li, 2025), FlashMLA-ETAP (Dege et al., 2025), and ThunderMLA (Spector et al., 2025) propose efficient kernel designs for MLA. Unfortunately, these kernels are agnostic to the shared prefix and an absorb-only implementation; therefore, they can not fully exploit hardware resources.

**Prefix Sharing:** Several prior works (e.g., SGLang, Hydragen etc.) have proposed techniques to exploit shared prefixes in attention calculations and KV-cache management (Juravsky et al., 2024; Zheng et al., 2024). Other frameworks (e.g., FlashInfer-Cascade, FastTree etc.) developed specialized MHA/GQA GPU kernels for tree-structured KV-caches to leverage the data reuse introduced by shared prefixes (Ye et al., 2024a; Zhu et al., 2024; Ye et al., 2024b; Yao et al., 2025; Pan et al., 2025; Wang et al., 2025b). However, these kernel designs target MHA/GQA architectures and aim to reduce HBM bandwidth usage. Consequently, they are not applicable to MLA, whose decode stage is typically compute-bound. In contrast, TyphoonMLA addresses this gap by reducing the number of FLOPS required in the compute-bound decode stage of MLA, achieving substantial speedups. Moreover, the mechanisms to handle complex tree-structured KV-caches introduced by the prior work would be equally applicable to both naive and absorb implementations. Therefore, applying these techniques to TyphoonMLA would be trivial.

## 6 CONCLUSIONS

In this paper, we proposed a novel MLA method, TyphoonMLA, which combines the naive and absorb implementations to effectively exploit shared prefixes in the KV-cache. Our theoretical analysis showed that, in the presence of a shared prefix, TyphoonMLA requires fewer FLOPS in compute-bound parts and smaller HBM bandwidth in memory-bound parts of the attention calculations than the existing MLA kernels. Experimental results on NPUs and GPUs show that TyphoonMLA improves attention throughput by up to 3.2× on DeepSeek-v3 and Kimi K2 models, while incurring only a minimal increase in HBM footprint.

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

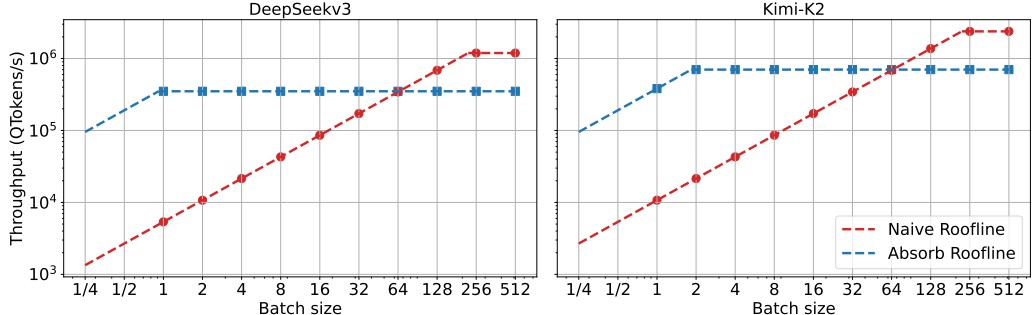

Figure 6: Roofline analysis of the naive and absorb implementations for DeepSeek-v3 and Kimi K2 models with a memory bandwidth of 1.8 TB/s and a cube throughput of 400 TFLOPS/s.

## A    APPENDIX

### A.1    ROOFLINE ANALYSIS

To analyze the computational characteristics of the naive and absorb implementations in the presence of a shared prefix, we plot the roofline model (Williams et al., 2009) of an AI accelerator for DeepSeek-v3 and Kimi K2 under both implementations, as shown in Fig.6. The y-axis in the plots represents throughput, calculated as the number of query tokens processed per second by the MLA kernel, while the x-axis represents the batch size, which determines the number of operations per byte (i.e., the operational intensity). The roofline model has two regions of operation: a bandwidth-limited region, where performance is limited by memory bandwidth due to insufficient operational intensity, and a compute-bound region, where performance is limited by the computational capacity of a processor, such as the total throughput of the cube units in NPUs and GPUs.

As expected, the absorb implementation provides higher throughput at low operational intensities thanks to the reduced memory bandwidth usage resulting from the compressed KV-cache stored in latent space. As batch size increases and operational intensity rises, the throughput of the absorb implementation does not improve further. For DeepSeek-v3, performance remains flat once the compute-bound region is reached, while for Kimi K2, throughput quickly saturates beyond a batch size of two. Thus, increasing operational intensity does not improve the performance of absorb implementations, as they are typically compute-bound.

In contrast, the naive implementation greatly benefits from the increasing operational intensity and achieves higher throughputs, as it is typically memory-bound at small batch sizes due to the uncompressed KV-cache. At batch sizes larger than 64, where significant data reuse occurs, the naive implementation achieves up to 3.4× higher throughput than the absorb implementation, thanks to its lower number of required operations. This analysis indicates that, although the absorb implementation is advantageous at low operational intensities, the naive implementation offers superior performance when sufficient operational intensity is present. These observations motivate our proposed kernel, which combines both approaches to maximize efficiency and performance across both compute and memory-bound regions of MLA calculations.

### A.2    THEORETICAL ANALYSIS OF TYPHOONMLA

In this section, we visualize the computational characteristics of the naive and absorb formulations using the computational model presented in Table 1. Fig. 7 depicts the estimated execution time of the absorb implementation, naive implementation, and TyphoonMLA for varying batch sizes.

In the shared context part of the attention calculations, the execution time of the absorb formulation increases linearly with the batch size, since its execution is compute-bound. In contrast, the execution time of the naive formulation remains constant until the batch size of 128, since its execution is memory-bound and the memory bandwidth usage does not change with the batch size in the shared

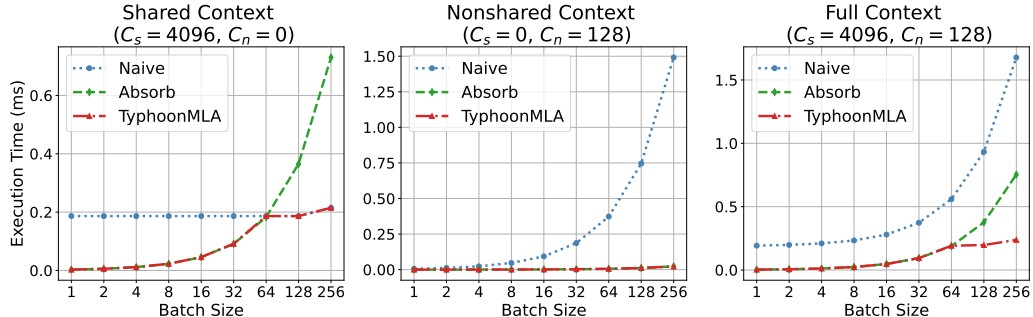

Figure 7: Theoretical analysis of Naive, Absorb, and TyphoonMLA.

prefix part of the KV-cache. Therefore, although faster at small batch sizes, the absorb becomes slower than the naive at batch sizes larger than 64. Therefore, TyphoonMLA switches from absorb to naive implementation at the batch size of 64 in order to achieve minimum execution time over the full range of batch sizes.

In the non-shared context part of the attention calculations, the absorb is always faster than naive. Therefore, TyphoonMLA employs the absorb implementation at all batch sizes. As a result of the combination of the two stages, TyphoonMLA is identical to the absorb implementation at batch sizes lower or equal to 64, beyond which the computational benefits of the naive implementation weighs in and makes TyphoonMLA significantly faster than the absorb baseline.

## A.3 SENSITIVITY TO BATCH SIZE

To analyze how different components of TyphoonMLA benefit from data reuse, we profile Typhoon-MLA and the baseline methods on an Ascend NPU across a range of batch sizes during the execution of DeepSeek-v3. Fig.8 shows the execution time of the shared part, non-shared part, and the overall attention calculation. Consistent with the complexity analysis in Section 3.2, the absorb baseline outperforms the naive baseline at small batch sizes in the shared part of the attention calculations due to the reduced memory bandwidth requirements of the compressed KV-cache in latent space, as shown in Fig.8(a). As the batch size increases, the execution time of the absorb baselines grows nearly linearly, while the naive baseline remains about the same, thanks to improved operational intensity. Around a batch size of 64, the naive baseline becomes faster than the absorb. To maintain

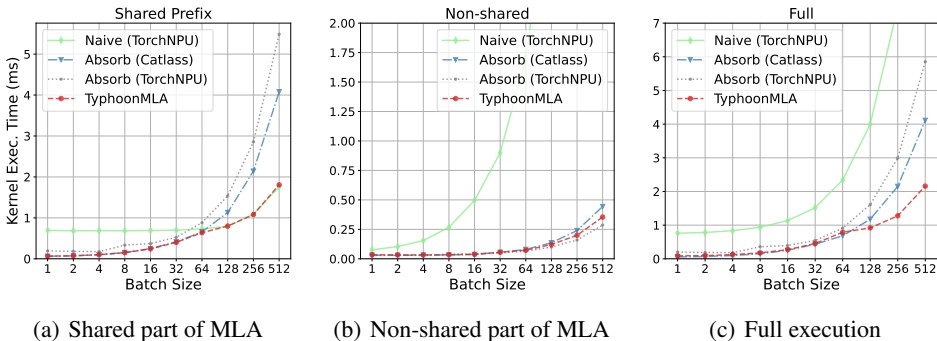

| (a) Shared part of MLA | (b) Non-shared part of MLA | (c) Full execution |

Figure 8: Performance breakdown of TyphoonMLA and the baselines on Ascend NPUs for DeepSeek-v3, assuming a shared prefix of length 4096 and a query length of 128. Execution time of individual components is measured using the CANN toolkit's msprof tool.

optimal performance across all batch sizes, TyphoonMLA switches from the absorb-only implementation to the mixed naive-absorb kernel once the batch size exceeds this cut-off point.

Fig.8(b) shows the execution time of TyphoonMLA and the baseline methods for the non-shared part of the attention calculations. Since this part does not contain any shared prefix, increasing the batch size does not improve the operational intensity; hence, the absorb implementations remain faster than the naive across all batch sizes. Consequently, TyphoonMLA consistently employs the absorb implementation for the non-shared part. When these two portions are combined, as shown in Fig.8(c), TyphoonMLA behaves identically to the absorb baseline up to the cut-off batch size of 64, beyond which TyphoonMLA outperforms the absorb baseline, achieving a speedup of up to $2\times$ at the batch size of 512. These results confirm that TyphoonMLA's performance gains primarily arise from exploiting the shared portion of the KV-cache.

