# OpenReview forum: "TyphoonMLA: A Mixed Naive-Absorb MLA Kernel For Shared Prefix"
_ICLR.cc/2026/Conference — ICLR 2026 Poster_

### Official Review · Reviewer_UutA · 2025-10-27

**Soundness:** 3
**Presentation:** 3
**Contribution:** 3
**Rating:** 4
**Confidence:** 3

**Summary:**

This paper introduces TyphoonMLA, a novel hybrid attention kernel designed to accelerate inference with shared prefixes for Large Language Models (LLMs) that utilize the Multi-Head Latent Attention (MLA) architecture, such as DeepSeek-v3 and Kimi K2. The authors identify a key performance bottleneck in existing MLA implementations (e.g., FlashMLA), which exclusively use a compute-bound "absorb" formulation to minimize HBM bandwidth. While effective for single sequences, this approach does not take advantage of data reuse opportunities in common shared-prefix scenarios (e.g., system prompts, tree-based decoding).

The core contribution is a hybrid method that partitions the attention calculation. It applies the compute-efficient "naive" formulation to the shared-prefix portion of the KV-cache, where high data reuse mitigates its higher bandwidth cost. Concurrently, it uses the bandwidth-efficient "absorb" formulation for the non-shared portions. By combining the outputs of these two parallel paths, TyphoonMLA achieves significant throughput gains—up to 3.24x on GPUs and 3x on NPUs—with a negligible HBM footprint overhead of approximately 3%.

**Strengths:**

1. The idea of creating a hybrid kernel that combines the naive and absorb formulations within a single attention computation is simple and insightful. Prior work has explored prefix sharing for standard MHA/GQA, but those mechanisms are typically memory-bound. This paper clearly identifies that MLA's decoding stage is compute-bound, making the problem fundamentally different and requiring a new approach. The solution is an effective combination of existing concepts tailored to the characteristics of the MLA architecture.
2. The Roofline analysis (Figure 6) clearly motivates the problem and visualizes the performance trade-off between the two formulations. The computational analysis in Table 1 provides a solid mathematical foundation for the expected gains. The experiments are thorough, validating the approach on two different hardware platforms (NVIDIA GPUs, Huawei Ascend NPUs), two state-of-the-art models, and realistic workloads using real-world system prompts of varying lengths. The inclusion of a latency breakdown (Figure 4) is a major strength, as it serves as a microbenchmark that directly validates the core hypothesis—that the speedup mainly comes from accelerating the shared part of the computation.
3. The paper is clearly written and easy to follow. Figure 1 provides a good visual schematic that contrasts the naive, absorb, and the proposed TyphoonMLA data flows, making the core mechanism immediately understandable. The results are presented clearly and support the paper's claims.
4. As LLMs are increasingly deployed with long and complex system prompts, and with the rise of reasoning techniques like Tree-of-Thought, optimizing for shared prefixes is becoming more important. The MLA architecture is used by several open-source models, and this work provides a solution that offers substantial, real-world performance improvements. The fact that it requires no model retraining and can be integrated into existing serving frameworks like vLLM increases its practical relevance.

**Weaknesses:**

1. The paper addresses the MLA kernel performance in shared prefix scenarios, but in Table 2, the system prompts are sampled from Claude-4, OpenAI/o3, and Grok/Personas, which do not use MLA. However, the models tested are DeepSeek-V3 and Kimi K2, which do use MLA. The experimental design could be improved by using system prompts that are more representative of DeepSeek-V3 and Kimi K2.
2. The paper does not explore how the length of the shared prefix affects acceleration. It is unclear at what length positive gains in TTFT and TGR metrics are achieved.
3. Section 3.1 describes the "Fall-back to Absorb" mechanism, but does not clearly provide the predefined threshold of gbs >= 64 in the main text or experiments; instead, it is discussed in the appendix. It is suggested to clarify this in the main paper in future versions.
4. The overhead of the "CombineLSE" step, which merges the results from the two parallel paths, is reported as small but is not analyzed. The paper would be improved by briefly discussing this step's computational complexity and how it scales with batch size or the number of attention heads, as this is important for understanding limitations in extreme-scale scenarios.
5. The authors claim the method is numerically identical to standard MLA, which is theoretically sound. However, the paper does not provide empirical evidence, such as a validation table confirming identical outputs (e.g., bit-for-bit) or benchmark results compared to the baseline. Such results would confirm the implementation's correctness.
6. The paper claims compatibility with systems like vLLM but does not sufficiently discuss the engineering complexity of deploying this method in dynamic, real-world serving environments. Key challenges are not addressed, such as managing a hybrid KV-cache layout within a memory manager like PagedAttention (which could increase fragmentation), or handling load imbalances due to highly variable non-shared sequence lengths within a batch.

**Questions:**

1. Table 1 and Section 3.2 do not clearly define what `k` means, for example in $L_s40k + BL_n40k$. Please clarify the meaning of `k`.
2. Figure 6 analyzes the Roofline for different batch sizes, but Figures 2 and 3 only show results for bsz=256, 512, and 1024. Could you add results for batch sizes 1, 8, 32, and 128? If small batch sizes are not relevant in practice, please provide supporting evidence.
3. Figure 3 shows only bsz=1024; can you also show results for multiple batch sizes on the GPU, similar to Figure 2 for the NPU?
4. Can the paper provide end-to-end acceleration comparisons for TTFT and TGR between FlashMLA, FlashInfer, and TyphoonMLA?
5. Could you elaborate on the system's behavior in a dynamic serving environment with heterogeneous requests? Specifically:
   a) How does the system handle load balancing between the naive and absorb paths when non-shared sequence lengths are highly variable within a batch?
   b) What are the specific memory management challenges (e.g., fragmentation) when integrating the hybrid KV-cache with continuous batching?
6. The performance gains are shown for a contiguous shared prefix. Have you considered generalizing this hybrid approach to more complex, non-contiguous sharing patterns (e.g., shared data blocks in a RAG context), or the potential applicability of the core "hybrid compute/memory-bound" principle to other architectures like standard MHA/GQA as hardware evolves?

---

> ### Author Response · Authors · 2025-11-21
>
> **Part 1/3**
>
> **We deeply thank the reviewer for their time to review our paper and for providing valuable suggestions to improve it.** We are particularly encouraged by the reviewer's acknowledgment of the effectiveness of the proposed method, the thoroughness of our Roofline analysis and experiments, as well as the quality of our writing. We provide our answers to the weaknesses and questions posed by the reviewer below.
>
> &nbsp;
>
> ---
>
> > The paper addresses the MLA kernel performance in shared prefix scenarios, but in Table 2, the system prompts are sampled from Claude-4, OpenAI/o3, and Grok/Personas, which do not use MLA. However, the models tested are DeepSeek-V3 and Kimi K2, which do use MLA. The experimental design could be improved by using system prompts that are more representative of DeepSeek-V3 and Kimi K2.
>
> The system prompts used in LLM services typically represent the alignment policy, safety features, available tool/API invocations, and in some cases, the information after the cut-off date of the model. Therefore, the system prompts are generally decoupled from the underlying models. For example, an open-sourced model such as Deepseek-v3 can be deployed by two different companies with two completely different system prompts. Therefore, we argue that using the system prompts from DeepSeek or Moonshot-ai in the experiments would not be critical to the overall evaluation. Moreover, companies often keep their system prompts confidential to minimize security risks. As a result, only the system prompts that have been leaked through prompt injection attacks presented in the public repo [1] are available to us, and unfortunately, none of the available system prompts are from the services that are known to use Deepseek-v3 or Kimi K2.
> [1] https://github.com/asgeirtj/system_prompts_leaks/tree/main/Misc
>
> &nbsp;
>
> ---
>
> > The paper does not explore how the length of the shared prefix affects acceleration. It is unclear at what length positive gains in TTFT and TGR metrics are achieved.
>
> To evaluate the impact of the length of the shared prefix, we used three system prompts in our experiments with the lengths of 26472, 7069, and 4759 taken from popular inference services  (described in Table 2). However, we agree that showing the minimum length that provides a speedup would be an important contribution to the evaluation. Therefore, we plan to run experimental results with artificial system prompt lengths and add them to the paper before the end of the rebuttal phase.
>
> &nbsp;
>
> ---
>
> > Section 3.1 describes the "Fall-back to Absorb" mechanism, but does not clearly provide the predefined threshold of gbs >= 64 in the main text or experiments; instead, it is discussed in the appendix. It is suggested to clarify this in the main paper in future versions.
>
> Before we submitted the paper, we had identified the batch threshold as 64 empirically by sweeping the batch size and comparing the throughput. However, we agree that this might not be a practical way of finding the batch size threshold. To that end, we derived a mathematical closed-form expression that finds this threshold as a function of architectural parameters and hardware specs based on the computational analysis provided in Section 3.2. When we substituted the parameters in the expression with those of Deepseekv3 and Ascend NPU specs, we found the threshold value as 61, which is quite close to the empirical value that we had found before.  We added this derivation in Section 3.2, so that anybody can easily find this threshold for any LLM architecture and any target hardware platform.
>
> &nbsp;
>
> ---
>
> > The overhead of the "CombineLSE" step, which merges the results from the two parallel paths, is reported as small but is not analyzed. The paper would be improved by briefly discussing this step's computational complexity and how it scales with batch size or the number of attention heads, as this is important for understanding limitations in extreme-scale scenarios.
>
> The number of operations and memory reads required by CombineLSE is proportional to the batch size, number of attention heads, and the hidden size. However, different from the other parts of the attention computations, it is independent of the KV sequence length, which makes its computational complexity insignificant compared to the other components of attention kernels. The computational overhead of the CombineLSE function can also be seen in the latency breakdown provided in Figure 4, which is less than 6% even at a batch size of 1024. To clarify the computational overhead of this function, we added a computational analysis for CombineLSE in Section 3.2 and discussed how it differs from the other parts of the attention calculations.

---

> > ### Author Response · Authors · 2025-11-21
> >
> > **Part 2/3**
> >
> > > The authors claim the method is numerically identical to standard MLA, which is theoretically sound. However, the paper does not provide empirical evidence, such as a validation table confirming identical outputs (e.g., bit-for-bit) or benchmark results compared to the baseline. Such results would confirm the implementation's correctness.
> >
> > To confirm the exactness of the proposed method, we implemented unit tests that execute the baseline flashinfer MLA function and the proposed kernel with the same input and calculate the mean and max. error in the outputs, asserting that the error is within an acceptable margin (any two functionally-equivalent but different implementations (e.g., flash-attention and flashinfer) would have a small difference in their outputs due to the undeterministic nature of the FP calculations on GPUs). Moreover, we provided these unit tests as supplementary material to this submission so that the members of the community can run and validate the correctness of the proposed method. We argue that validating our method's correctness through open-sourced code and tests would be more valuable than claiming it in the paper, while it would also save valuable space in the manuscript.
> >
> > &nbsp;
> >
> > ---
> >
> > > Table 1 and Section 3.2 do not clearly define what k means. Please clarify the meaning of k.
> >
> > k in Table 1 and Section 3.2 was meant to represent the binary kilo unit (=1024). However, we agree that this usage of k in the table creates confusion. Therefore, we scaled the right-most column in Table 1 by 1024 and removed k from the expressions.
> >
> > &nbsp;
> >
> > ---
> >
> > > Figure 6 analyzes the Roofline for different batch sizes, but Figures 2 and 3 only show results for bsz=256, 512, and 1024. Could you add results for batch sizes 1, 8, 32, and 128?
> >
> > Since there is not sufficient data reuse at small batch sizes, TyphoonMLA falls back to its absorb-only implementation when the batch size is equal to or smaller than 64 (Section 3.1), and its performance becomes equal to the CATLASS-absorb baseline. Therefore, we omit the results with batch sizes equal to or smaller than 64.
> >
> > &nbsp;
> >
> > ---
> >
> > > If small batch sizes are not relevant in practice, please provide supporting evidence.
> >
> > While a small batch size might be required for certain applications with tight SLO constraints or limited HBM capacity, large batch sizes typically increase the computational efficiency and improve the cost per token. Therefore, many online LLM services employ batched inference with large batch sizes to improve efficiency and cut their cost. For instance, Pope et al. suggest a batch size of 512 for PaLM 540B on TPU clusters when high-throughput is prioritized [1]. Therefore, we believe that improving the computational efficiency of LLM inference with the proposed method would contribute to many deployment scenarios where computational efficiency and cost reduction are prioritized.
> > [1] R. Pope et al., "Efficiently scaling transformer inference." Proceedings of machine learning and systems (MLSys). 2023.
> >
> > &nbsp;
> >
> > ---
> >
> > > Figure 3 shows only bsz=1024; can you also show results for multiple batch sizes on the GPU, similar to Figure 2 for the NPU?
> >
> > We repeated our GPU experiments for batch sizes 128, 256, and 512 and added them to Figure 3. For completeness, we also added the NPU experimental results with bsz=128 to the Fig. 2.
> >
> > &nbsp;
> >
> > ---
> >
> > > Can the paper provide end-to-end acceleration comparisons for TTFT and TGR between FlashMLA, FlashInfer, and TyphoonMLA?
> >
> > Our method targets only the self-attention layers in LLMs; therefore, TyphoonMLA does not affect the performance of other operations, such as linear projections, FFN, or MoE layers. Consequently, the end-to-end acceleration results would depend on the ratio of the execution time of self-attention to the total time. By using the profiling data from a cluster of 128 H100 GPUs provided by Deepseek [1] and scaling the self-attention time to the sequence lengths required by the system prompts and datasets used in our experiments (max. seq. len: 9k - 30k), we estimate that the ratio of self-attention to total time would be between 0.56 and 0.8. As a result, the reported speedups obtained in Fig. 3 would translate into an end-to-end acceleration in terms of TGR by a reduction between 20% - 44%. For example, for the Deepseek-v3 model with System Prompt A and the GSM8K dataset with a batch size of 128, TyphoonMLA would achieve a 1.35x end-to-end speedup in terms of TGR.
> > As the proposed method addresses only decode, not prefill. Therefore, TyphoonMLA does not have any impact on the TTFT.
> > [1] https://github.com/deepseek-ai/profile-data

---

> > > ### Author Response · Authors · 2025-11-21
> > >
> > > **Part 3/3**
> > >
> > > > Could you elaborate on the system's behavior in a dynamic serving environment with heterogeneous requests? Specifically:
> > > a) How does the system handle load balancing between the naive and absorb paths when non-shared sequence lengths are highly variable within a batch?
> > > b) What are the specific memory management challenges (e.g., fragmentation) when integrating the hybrid KV-cache with continuous batching?
> > >
> > > Our CATLASS kernel for the non-shared part takes an auxiliary input with the non-shared KV sequence length of each request and performs attention computation accordingly. The KV-sequence dimension of each request is partitioned into chunks of 128 rows and distributed across the 24 cores of NPUs. As a result of the variable sequence lengths, some requests require fewer chunks to be processed by the cores. However, when a core finishes processing a chunk of a request, it proceeds to process the chunks of other requests, minimizing the idle time. As a result, the proposed kernel implementation maintains high hardware utilization even when processing a batch with highly variable KV-sequence lengths. As the shared part of the KV-cache has the same sequence length across the requests, the computation of the naive part is equally distributed across the cores.
> > >
> > > To avoid memory fragmentation problems in KV-cache management, we adopt paged attention and partition the KV-cache into smaller, fixed-sized chunks, which is a technique widely adopted by popular inference frameworks such as vLLM and SGLang. As a result, we did not experience any fundamental or significant issues with KV-cache management when building the proposed system.
> > >
> > > &nbsp;
> > >
> > > ---
> > >
> > > > The performance gains are shown for a contiguous shared prefix. Have you considered generalizing this hybrid approach to more complex, non-contiguous sharing patterns (e.g., shared data blocks in a RAG context), or the potential applicability of the core "hybrid compute/memory-bound" principle to other architectures like standard MHA/GQA as hardware evolves?
> > >
> > > This paper primarily focuses on shared prefix due to its widespread usage in LLM serving systems; however, this method can be equivalently extended to more complex shared KV-cache structures, such as a radix tree or shared data blocks. We consider this direction as future work for this paper.
> > >
> > > In the standard MHA/GQA, there is no mechanism similar to the absorption trick in MLA that allows us to easily trade compute with memory bandwidth. Therefore, we do not see a straightforward way of applying the same idea to the MHA/GQA architectures.
> > >
> > > &nbsp;
> > >
> > > ---
> > >
> > > **In summary, following the feedback from this review, we made the following changes to the paper:**
> > > 1. Performed additional experimental results with a batch size of 128 on NPUs and 128, 256, and 512 on GPUs, and updated Figure 2 and 3.
> > > 2. Derived a mathematical closed-form to identify the batch size threshold analytically and added it in Section 3.2.
> > > 3. Added a paragraph to discuss the computational overhead of the CombineLSE function in Section 3.2.
> > > 4. Simplied the expressions and visually improved Table 1.
> > >
> > > **We thank the reviewer once again for their time and valuable feedback.** We believe the revisions and additional experiments have substantially strengthened the paper, and we hope our responses adequately address all concerns. If any part of our clarification is unclear or if further questions arise, we would be glad to provide additional details. If you find that the improvements meaningfully enhance the quality of our paper, we would greatly appreciate your reconsideration of the review score in light of these updates.

---

> > > > ### Comment · Reviewer_UutA · 2025-11-25
> > > >
> > > > Thank you for the authors’ response, which has addressed most of my concerns and incorporated several suggestions, leading to meaningful improvements in the manuscript. To further enhance the quality of the paper, I have several additional recommendations:
> > > >
> > > > > To that end, we derived a mathematical closed-form expression that finds this threshold as a function of architectural parameters and hardware specs based on the computational analysis provided in Section 3.2. When we substituted the parameters in the expression with those of Deepseekv3 and Ascend NPU specs, we found the threshold value as 61, which is quite close to the empirical value that we had found before.
> > > >
> > > > I appreciate the authors’ derivation of the mathematical closed-form expression for the predefined threshold of GBS and the empirical validation that identifies the optimal threshold as 61, which is very close to the previous version’s value of 64. To facilitate clearer comparison, I suggest including results for gbs=64 in Figures 2 and 3. Furthermore, for better visual clarity, it would be beneficial to group the bars corresponding to the same GBS values together.
> > > >
> > > >
> > > > > The authors claim the method is numerically identical to standard MLA
> > > >
> > > > The authors have provided unit test results verifying correctness. However, the equivalence is mathematical rather than strictly numerical. I therefore recommend revising this description to accurately reflect mathematical equivalence.
> > > >
> > > >
> > > > > TyphoonMLA would achieve a 1.35x end-to-end speedup in terms of TGR
> > > >
> > > > I suggest including this finding in the Abstract. Although the primary focus of the paper is on the Attention module, with reported improvements of 3× and 3.24× speedup on NPUs and GPUs respectively, the end-to-end acceleration is of particular practical interest to the industry. Highlighting this in the Abstract will help readers quickly grasp the real-world impact of your method.
> > > >
> > > > **I hope these suggestions help further refine your manuscript and improve its clarity and impact.**

---

> > > > > ### Author Response · Authors · 2025-11-27
> > > > >
> > > > > We are happy to hear that our previous responses have addressed most of your initial concerns and the revisions have led to meaningful improvements in the paper. **We sincerely thank you for your constructive feedback and insightful suggestions.**
> > > > >
> > > > > > I appreciate the authors’ derivation of the mathematical closed-form expression for the predefined threshold of GBS and the empirical validation that identifies the optimal threshold as 61, which is very close to the previous version’s value of 64. To facilitate clearer comparison, I suggest including results for gbs=64 in Figures 2 and 3. Furthermore, for better visual clarity, it would be beneficial to group the bars corresponding to the same GBS values together.
> > > > >
> > > > > We repeated the experiments for bsz=64 on the GPU and added them in Figure 3. Following your suggestion, we have restructured the plot layout and grouped the bars by the batch size. We believe that the plots are now much easier to read, and performance gains are much more visible to the reader. The NPU experiments for bsz=64 are still running, and we will add them to Figure 2 as soon as they are completed.
> > > > >
> > > > > > The authors have provided unit test results verifying correctness. However, the equivalence is mathematical rather than strictly numerical. I therefore recommend revising this description to accurately reflect mathematical equivalence.
> > > > >
> > > > > We agree that it is more correct to say that TyphoonMLA is mathematically equivalent to the standard MHA. Therefore, we rephrased the sentences in L84 and L239, emphasizing that the method is mathematically equivalent rather than numerically identical.
> > > > >
> > > > > > I suggest including this finding in the Abstract. Although the primary focus of the paper is on the Attention module, with reported improvements of 3× and 3.24× speedup on NPUs and GPUs respectively, the end-to-end acceleration is of particular practical interest to the industry. Highlighting this in the Abstract will help readers quickly grasp the real-world impact of your method.
> > > > >
> > > > > To incorporate these end-to-end results in the paper, we added a new subsection, **End-to-End Speedup**, to the Experiments section and described our methodology on how we obtained the TGR numbers. Then, we added Table 3, which shows the attention time, per-iteration end-to-end decode time, and TGR for TyphoonMLA and FlashMLA for three system prompts. Following your suggestion, we also added the end-to-end speedup numbers in terms of TGR in the abstract.
> > > > >
> > > > > **We thank you again for these suggestions, which substantially improved the quality and clarity of our submission.** Should you believe that our submission has improved since the beginning of the rebuttal period, we kindly ask you to reconsider your review score.

---

> > > > > > ### Comment · Reviewer_UutA · 2025-11-27
> > > > > >
> > > > > > Thank you for the detailed response. The authors have effectively addressed my concerns. Accordingly, I am increasing my score from 4 to 6.

---

> > > > > > > ### Author Response · Authors · 2025-11-27
> > > > > > >
> > > > > > > We appreciate your reconsideration of the review score. We thank you once again for your time and the fruitful and constructive technical discussions.

---

### Official Review · Reviewer_tmtF · 2025-10-28

**Soundness:** 3
**Presentation:** 3
**Contribution:** 2
**Rating:** 6
**Confidence:** 3

**Summary:**

This paper addresses a performance bottleneck in modern Large Language Models (LLMs) that use Multi-Head Latent Attention (MLA), such as DeepSeek-v3. The standard absorb implementation used for decoding is memory-efficient but compute-bound, preventing it from benefiting from data reuse in scenarios with shared prefixes (e.g., system prompts).

The authors propose TyphoonMLA, a novel hybrid attention kernel that partitions the computation. It applies the computationally cheaper naive formulation to the shared prefix portion of the Key-Value Cache to reduce floating-point operations (FLOPs), while using the memory-bandwidth-efficient absorb formulation for the non-shared parts. This hybrid approach is a bit-exact, drop-in replacement that requires no model retraining.

**Strengths:**

While prefix sharing optimization isn't new, the paper is the first to identify and address the specific bottleneck it creates for Multi-Head Latent Attention (MLA) models.

The computational analysis in Section 3.2 (Table 1) is clear and compelling. It precisely breaks down the HBM read and MAC requirements, mathematically demonstrating why the hybrid approach should be superior by reducing MACs in the compute-bound shared region and HBM reads in the memory-bound non-shared region.

Besides, this is not a theoretical optimization for a toy problem. It directly accelerates models that are at the forefront of LLM development (DeepSeek, Kimi). As system prompts and context lengths grow, this bottleneck becomes increasingly important.

**Weaknesses:**

My main issue with the paper is that the core idea feels less novel than presented. Splitting attention into shared and non-shared parts is a known trick in the MHA/GQA world—we've seen this before in libraries like FlashInfer with its "Cascade Inference" and in Hydragen. The paper doesn't acknowledge this prior art, which makes its contribution seem more like a clever adaptation to MLA's specific compute/memory tradeoffs rather than a brand-new algorithm.

Beyond that, there are a few other points that need tightening up:
1. The paper is a bit hand-wavy about the "predefined threshold" used to switch back to the absorb kernel. It's not clear how this is set or how sensitive the performance is to this magic number.

2. It claims to handle complex tree-structured prefixes (like in Tree-of-Thought), but doesn't show how. This is a non-trivial problem that other kernels, like FlashForge, were specifically designed to solve.

3. The NPU results are impressive, but they feel a bit like a black box. Because the implementation relies on proprietary Huawei libraries (CANN/CATLASS), it's hard to judge how portable the approach is. The paper doesn't discuss the underlying principles that would make this work on other NPU architectures

**Questions:**

You mention that the kernel falls back to an absorb-only implementation below a "predefined threshold". How is this threshold determined? Is it found empirically for each hardware/model pair? It would be helpful to understand how sensitive the overall performance is to this value. An ablation study on this threshold would add a lot of value.

You claim compatibility with tree-structured workloads like Tree-of-Thought. However, the method described seems tailored to a simple linear prefix. How would TyphoonMLA handle a more complex KV cache layout, where there might be a root, several shared branches, and then unique leaves?

---

> ### Author Response · Authors · 2025-11-21
>
> **Part 1/2:**
>
> **We sincerely thank the reviewer for their time and constructive feedback.** We are glad to hear that the reviewer finds our computational complexity analysis useful and recognizes the importance of the problem that this paper addresses. Below, we address the specific concerns raised in this review and explain the changes that we made in the paper accordingly.
>
> &nbsp;
>
> ---
>
> > You mention that the kernel falls back to an absorb-only implementation below a "predefined threshold". How is this threshold determined? Is it found empirically for each hardware/model pair? It would be helpful to understand how sensitive the overall performance is to this value. An ablation study on this threshold would add a lot of value.
>
> Before the initial submission, we had empirically identified the predefined threshold as 64 by measuring performance on NPUs for varying batch sizes. However, we agree that this is not an optimal way of finding this threshold value. Therefore, we derived a mathematical closed-form expression in terms of the architectural parameters and hardware specifications based on our computational analysis presented in Section 3.2. When we substitute the parameters with Deepseek-v3 parameters and Ascend NPU specs, we calculate the threshold value as 61, which is quite close to the empirical value that we found earlier. This way, any user of TyphoonMLA can easily identify this threshold for any architecture and target hardware platform. We added our derivation and the mathematical closed-form expression to Section 3.2.
>
> &nbsp;
>
> ---
>
> > The NPU results are impressive, but they feel a bit like a black box. Because the implementation relies on proprietary Huawei libraries (CANN/CATLASS), it's hard to judge how portable the approach is. The paper doesn't discuss the underlying principles that would make this work on other NPU architectures
>
> In order to show that the proposed method is portable across different hardware architectures, we implemented TyphoonMLA on GPUs and shared the experimental results from GPUs in Figure 3. In the original submission, however, Figure 3 showed only a batch size of 1024. For a better comparison with NPU results, we performed additional experiments on GPUs with batch sizes of 128, 256, and 512 during the rebuttal period and added the results in Figure 3. We observe that TyphoonMLA achieves speedups on GPUs similar to NPUs (up to 3.24x), which demonstrates that the performance gains obtained by the proposed method are not limited to the Ascend NPUs, but generalize to other architectures such as GPUs. More information about our GPU experiments is available in Section 4, subsection "Experiments on GPUs".
>
> &nbsp;
>
> ---
>
> > My main issue with the paper is that the core idea feels less novel than presented. Splitting attention into shared and non-shared parts is a known trick in the MHA/GQA world—we've seen this before in libraries like FlashInfer with its "Cascade Inference" and in Hydragen. The paper doesn't acknowledge this prior art, which makes its contribution seem more like a clever adaptation to MLA's specific compute/memory tradeoffs rather than a brand-new algorithm.
>
> We agree that the novelty of this paper stems from applying the concepts introduced in the Hydragen and Flashinfer-Cascade papers to MLA models, by rethinking how we can effectively apply those techniques while accounting for the unique computational characteristics of MLA architectures. Although we had already cited and discussed both FlashInfer-Cascade and Hydragen in the original submission (L51, L464, L466), we changed the wording in Section 5 in the new revision to emphasize this prior art and make them more visible to the reader.
>
> &nbsp;
>
> ---
>
> > You claim compatibility with tree-structured workloads like Tree-of-Thought. However, the method described seems tailored to a simple linear prefix. How would TyphoonMLA handle a more complex KV cache layout, where there might be a root, several shared branches, and then unique leaves?
>
> In this paper, we focused on shared prefix due to its widespread adoption in LLM serving systems in the form of system prompts. We agree that implementing more complex tree structures is non-trivial, but we argue that the same mechanisms introduced by the prior work, such as FlashForge or FlashInfer-Cascade, are equally applicable to both naive and absorb implementations in MLA architectures. As a result, applying the same mechanisms to the naive and absorb components of our TyphoonMLA implementation would enable supporting more complex tree-structured KV caches. We added this description to Section 5, and we consider implementing these features and evaluating them in various application domains (e.g., tree-of-thought, speculative decoding) as future work.
>
> **continues in the next comment...**

---

> ### Author Response · Authors · 2025-11-21
>
> **Part 2/2:**
>
> In summary, based on your feedback, we made the following changes in the paper, which can be seen in the new revision:
> 1. We conducted additional experiments with batch sizes of 128, 256, and 512 on GPUs (Figure 3) to facilitate a better and clearer comparison to NPUs.
> 2. We derived a mathematical closed-form expression to calculate the batch size threshold in terms of architectural parameters and hardware specs (Section 3.2).
> 3. We described how a more complex tree-structured KV cache could be applied to TyphoonMLA in Section 5.
> 4. We improved the wording to make prior work (Hydragen and FlashInfer-Cascade) stand out to the reader.
>
> **We thank the reviewer again for their constructive feedback.** We believe that these changes significantly strengthened our submission, and we hope that our responses address the concerns raised in this review. If you require any clarification on any of our responses above or have any further feedback, we would be happy to address them. Should you believe that the additional experiments and the modifications to the manuscript improved the quality of this paper, we would be grateful for a reconsideration of the review score.

---

> > ### Comment · Reviewer_tmtF · 2025-11-26
> >
> > I appreciate the authors' detailed response and the effort put into the additional experiments. The new closed-form derivation for the switching threshold is a significant improvement over the previous empirical approach, and the added GPU experiments with varying batch sizes effectively address my concerns about the method's portability beyond specific NPU libraries.
> >
> > That said, I am inclined to maintain my original score of 6. While these revisions have certainly solidified the paper's quality and completeness, my assessment regarding the fundamental novelty remains unchanged. I believe the current rating fairly reflects the paper as a solid, acceptance-worthy engineering contribution.
> >
> > I will increase the soundness score and my confidence score.

---

> > > ### Author Response · Authors · 2025-11-27
> > >
> > > We are pleased to hear that the reviewer considers our new closed-form derivation and the additional GPU experiments as a significant improvement. We thank the reviewer once again for their time and constructive feedback.

---

### Official Review · Reviewer_jvoP · 2025-10-31

**Soundness:** 4
**Presentation:** 3
**Contribution:** 4
**Rating:** 8
**Confidence:** 4

**Summary:**

This paper presents a method that accelerates inference for transformers in the presence of shared prefixes, such as system prompts in modern LLMs. The method, called TyphonMLA, is a hybrid kernel that combines regular attention and MLA from DeepSeek. It consists of using regular attention for shared prefixes and MLA for the rest. Moreover, the kernel falls back to standard MLA at small batch sizes. In practice, the hybridization is carried by merging both attention results via a shared softmax denominator. On NPUs and GPUs, with real long system prompts (Claude 4, GPT o3, Grok), TyphoonMLA reaches up to 3x throughput over standard FlashMLA baselines, with just 3% HBM overhead.

**Strengths:**

There are many things that I liked in this paper and that I believe it will be helpful to others at ICLR:

-  MLA decode today is compute-bound because of absorb, so prefix-sharing kernels from MHA/GQA don't directly help. TyphonMLA fills this gap.

- The idea is very simple but impactful.

- The paper brings concrete computational analysis. For example, Table 1 shows MAC ops and HBM reads for naive, absorb, and TyphoonMLA; and Figure 4 shows a detailed latency breakdown of TyphonMLA across different stages of the kernel.

- Since the idea is quite straightforward and the method is exact (same output as original MLA), TyphonMLA works well with standard inference techniques, such as paged KV. Therefore, it can be easily included into frameworks like vLLM and SGLang.

- Strong results on both NPUs and GPUs, with two real MLA models (DeepSeek-v3, Kimi K2) using multiple real system prompts.

**Weaknesses:**

A few things to be improved in the paper include:

- Right now Table 1 is overwhelming to read as it contains a lot of symbols. One idea is to organize the table by placing MAC rows first and HBM reads rows later so that methods are more easily comparable. Distributing $L_s$ and $L_n$ for all methods would also improve things. E.g., $BS_q (L_s + L_n) H (2D_l + D_r)$ to $BS_q L_s H(2D_l + D_r) + B S_q L_n H(2D_l + D_r)$.

- The impressive results of 3x speedup happens in shared prefix scenarios, which is not always the case in transformers.

- I believe the bar plots are clean and informative, but they would be much better with error bars / confidence intervals to get an idea of how robust are the results.

**Questions:**

Can you describe in more details how TyphonMLA kernel is implemented in a pseudocde?

---

> ### Author Response · Authors · 2025-11-21
>
> **Thank you very much for taking the time to review our paper and for providing valuable, constructive feedback.** We are encouraged by your positive assessment of the proposed method's impact, the soundness of our computational analysis and latency breakdowns, and the strength of our experimental results. Below, we provide our detailed responses to the concerns raised in your review.
>
> ---
>
> > Right now Table 1 is overwhelming to read as it contains a lot of symbols. One idea is to organize the table by placing MAC rows first and HBM reads rows later so that methods are more easily comparable. Distributing Ls and Ln for all methods would also improve things.
>
> We applied these suggestions in Table 1, and we believe that it is now easier to read and looks much better (you can see it in the new revision). Thanks for these suggestions, and please let us know if you have any further feedback.
>
> ---
>
> > I believe the bar plots are clean and informative, but they would be much better with error bars / confidence intervals to get an idea of how robust are the results.
>
> We agree that repeating each experiment multiple times and showing a confidence interval in the plot would improve the reliability of the results, but running all experiments for a single batch size takes about a day. Therefore, considering all batch sizes and both GPU and NPU figures, repeating all experiments for a statistically meaningful number of times is unfortunately prohibitively costly. However, the throughput values that we report in Fig.2 and 3 are the average throughput over thousands of iterations required to process an entire dataset; therefore, any variance across function calls is averaged out, and the final value is highly consistent across multiple runs. To confirm this claim, we repeated a single experiment 5 times with bsz=1024, model=DSv3, system_prompt=A, dataset=SimpleQA, and obtained the following throughput numbers:
>
> run1: 304.445 TeraOps/s
>
> run2: 304.307 TeraOps/s
>
> run3: 304.237 TeraOps/s
>
> run4: 304.486 TeraOps/s
>
> run5: 304.291 TeraOps/s
>
> These numbers support our claim that the throughput values reported in Fig.2 and 3 are highly consistent across multiple runs. Moreover, we open-sourced our implementation and experiment script (attached to the supplementary materials). Therefore, anyone in the community can reproduce our experiments and confirm that the experimental results are consistent across multiple runs.
>
> ---
>
> > Can you describe in more details how TyphonMLA kernel is implemented in a pseudocde?
>
> Following your suggestion and thanks to the one extra page allowance during and after the rebuttal period, we added Algorithm 1 in Section 3.1, which shows the details of TyphoonMLA. Please let us know if you have any feedback on it, and we would be happy to make necessary changes.
>
> ---
>
> **We thank you again for your valuable feedback and for helping us improve the quality of our paper.** We would be glad to discuss further or address any additional comments you may have.

---

> > ### Comment · Reviewer_jvoP · 2025-11-27
> >
> > I thank the authors for addressing my questions. I remain positive about this paper.

---

### Author Response · Authors · 2025-12-03
**Authors' Message to AC**

**We sincerely thank the reviewers and ACs for their time and effort**, especially under the extraordinary circumstances of this year's ICLR review process. The paper was positively received in the initial reviews, and during the rebuttal period, we substantially strengthened the work following the reviewers' suggestions. Below, we summarize the main strengths identified by the reviewers and the improvements made during the discussion period.

&nbsp;

### **Strengths initially identified by reviewers:**
* Addresses a fundamental performance bottleneck on MLA architectures (*jvoP, tmtF*)
* Simple, insightful, and impactful idea (*jvoP, UutA*)
* Applicability to real-world models and frameworks (*jvoP, tmtF, UutA*)
* Clear, concrete computational analysis (*jvoP, tmtF, UutA*)
* Thorough and strong experimental results on both NPUs and GPUs (*jvoP, tmtF, UutA*)
* Clear and easy-to-follow manuscript (*UutA*)

&nbsp;

### **Improvements made during the rebuttal phase:**
In response to the reviewers' requests, we provided the following additional experiments, analyses, and clarifications:
* New NPU experiments with batch sizes of 64 and 128 (Figure 2)
* New GPU experiments with batch sizes of 64, 128, 256, and 512 (Figure 3)
* Added Table 3 (Section 4) reporting end-to-end speedup using the TGR metric
* Added Algorithm 1 (Section 3.1) outlining the TyphoonMLA implementation
* Derived a closed-form analytical expression for the optimal batch size threshold (Section 3.2)
* Added an analysis of the computational overhead of the CombineLSE function (Section 3.2)
* Numerous improvements for clarity in text and figures (Figures 2, 3, Table 1, etc.)

&nbsp;

### **Reviewer responses during discussion:**
The additional experiments and revisions were well received by reviewers:
Reviewer jvoP (score: 8):
> "I thank the authors for addressing my questions. I remain **positive** about this paper."

Reviewer tmtF (score: 6):
> "The new closed-form derivation for the switching threshold is **a significant improvement** over the previous empirical approach, and the added GPU experiments with varying batch sizes **effectively address my concerns** about the method's portability beyond specific NPU libraries."

Reviewer UutA (score: 4 → 6):
> "...has **addressed most of my concerns** and incorporated several suggestions, leading to **meaningful improvements** in the manuscript.... The authors have **effectively addressed my concerns**. Accordingly, I am increasing my score from 4 to 6."

&nbsp;

**We thank the reviewers and ACs once again for their time and consideration.** We believe that the submission was already in a strong position at the beginning of the rebuttal period, and that it has been substantially improved thanks to the reviewers' invaluable suggestions. Before the openreview leak incident happened, reviewers unanimously recommended acceptance with scores of 8/6/6 and confidence 4/4/3. Given the strong initial evaluations and the substantial, well-received improvements made during the rebuttal period, we respectfully believe that this paper merits consideration for spotlight nomination.

---

### Meta-Review · Area_Chair_jkBB · 2025-12-24

**Summary:**

This paper proposes a hybrid MLA decoding kernel which uses naive mode (less compute, more HBM access but can be shared) to compute shared prefix and uses absorb mode to compute the remaining. This addresses an important issue of the original MLA, where the absorb mode is too computationally heavy. Descent speedup result is observed.

The reviewers have a consensus that the paper can be have useful in practice, despite the idea is very simple.

**Reviewer Concerns:**

Reviewer concerns include novelty (whether the proposed technique is a brand new contribution or such idea already showed in FlashInfer), transparentness of NPU results, soundness of using Claude/OpenAI/Grok prompts as shared prefix for DeepSeek, explicit TGR metrics, etc. Except the novelty, other issues are addressed during the rebuttal, leading to an improvement of the paper quality. Though the novelty might be limited, all the reviewers agree the impact should be good.

**Reviewer Scores:**

All the three authors are positive with the paper, with rating at (8,6,6).

---

### Decision · Program_Chairs · 2026-01-26

Accept (Poster)